# Effect of Magnetized Coagulants on Wastewater Treatment: Rice Starch and Chitosan Ratios Evaluation

**DOI:** 10.3390/polym14204342

**Published:** 2022-10-15

**Authors:** Nomthandazo Precious Sibiya, Gloria Amo-Duodu, Emmanuel Kweinor Tetteh, Sudesh Rathilal

**Affiliations:** Green Engineering Research Group, Department of Chemical Engineering, Faculty of Engineering and The Built Environment, Durban University of Technology, Durban 4001, South Africa

**Keywords:** coagulation, chitosan, rice starch, magnetite, wastewater treatment, magnetized coagulants

## Abstract

Coagulation with synthetic chemicals has been used to treat a wide range of industrial effluents. Herein, the unique characteristics of industrial effluents being detrimental to the environment warrants urgent resource-efficient and eco-friendly solutions. Therefore, the study investigated the use of two magnetized coagulants (chitosan magnetite (CF) and rice starch magnetite (RF)), prepared via co-precipitation in three different ratios (1:2, 1:1 and 2:1) of natural coagulants (chitosan or rice starch) and magnetite nanoparticles (F) as alternative coagulants to alum for the treatment of wastewater. A Brunauer–Emmett–Teller (BET) analyzer, an X-ray diffraction (XRD) analyzer, and energy-dispersive X-ray (EDX) spectroscopy were used to characterize the surface area, crystal structure, and elemental composition of the coagulants. The influences of settling time (10–60 min) on the reduction of turbidity, color, phosphate, and absorbance were studied. This was carried out with a jar test coupled with six beakers operated under coagulation conditions of rapid stirring (150 rpm) and gentle stirring (30 rpm). Wastewater with an initial concentration of 45.6 NTU turbidity, 315 Pt. Co color, 1.18 mg/L phosphate, 352 mg/L chemical oxygen demand (COD), and 73.4% absorbance was used. The RF with a ratio of 1:1 was found to be the best magnetized coagulant with over 80% contaminant removal and 90% absorbance. The treatability performance of RF (1:1) has clearly demonstrated that it is feasible for wastewater treatment.

## 1. Introduction

The demand on freshwater resource supply is deteriorating as industrial and agricultural activities are upsurging and population growth shows a constant increase. Consequentially, wastewater treatment becomes inevitable due to the detrimental effects it has on human health and the environment [1]. Water scarcity affects approximately 2 billion people [2]. Wastewater treatment plants (WWTPs) are a substantial source of pollutants discharged into water bodies [3,4] due to the fact that they do not meet the Environmental Protection Agency’s strict criteria for controlling effluent plant quality [5,6]. Therefore, it is critical for WWTPs to release clean effluent into water bodies since they serve as potable water and are utilized for agriculture purposes and other recreational activities. Various processes and technologies have been explored to enhance the quality of water in order to meet the water demand at a low cost [7]. These technologies are divided into three categories: physical (settling, filtration and membrane technology), chemical (coagulation, ion exchange, disinfection, oxidation, catalytic reduction and softening processes) and biological (microbial biodegradation, bioreactor processes, etc.) treatment techniques [8,9]. Coagulation/flocculation is frequently used in water and wastewater treatment to efficiently reduce the organic load prior to subsequent treatment processes [10]. Furthermore, it has sparked widespread attention in the industrial sector due to easy usability, high efficiency, and low cost. In coagulation, tiny particles are aggregated into bigger aggregates (flocs) and dissolved organic matter is adsorbed onto particulate aggregates so that these contaminants may be eliminated in later solid/liquid separation processes [11,12]. 

Coagulation involves a sequence of stages, beginning with electrostatic attraction between cationic proteins and negatively charged suspended contaminants, and ending with particle destabilization [11,13,14]. It occurs through three distinct mechanisms: (i) charge neutralization (particle destabilization at low coagulant dosage), (ii) sweep (the addition of coagulant at sufficiently high concentrations to produce anhydrous, amorphous precipitates enmeshing colloidal particles in these precipitates), and (iii) bridge formation [14,15]. Synthetic coagulants are classified as hydrolyzing metallic salts, pre-hydrolyzing metallic salts, and synthetic cationic polymers [16]. The most widely used coagulant salts are Al_2_(SO_4_)_3_, Fe_2_(SO_4_)_3_, AlCl_3_, and FeCl_3_ [1,17]. Although their efficacy has been well established, their usage is associated with high costs and environmental drawbacks due to their non-native origin, non-degradability, and the health problems linked to their leftover residues [18,19]. The excessive levels of aluminum tend to lower the pH of water and can accumulate in the food chain. Therefore, these drawbacks have resulted in a search for eco-friendly and sustainable coagulants in their use and availability. The progress of nanotechnology has substantially contributed to the creation of innovative ways for tackling many health and environmental concerns while using less energy [5]. 

Several studies have investigated the use of iron oxide nanoparticles, such as maghemite (y-Fe_2_O_3_), magnetite (Fe_3_O_4_) and hematite (∝-Fe_2_O_3_), in the removal of heavy metals and hazardous chemicals from water, demonstrating the effectiveness of these materials [20,21,22]. These are seen as very promising due to their multifunctional ability and physicochemical properties, which include superparamagnetic behaviors, smaller size, high surface area, environmental friendliness, and biodegradability [5,23]. Natural coagulants provide a cost-effective, environmentally acceptable, and long-term alternative to the use of synthetic chemicals [5,14]. They are generated or derived from a variety of sources, including microbes, animals, and plants [1]. Magnetic iron oxide nanoparticles functionalized with polymers such as eggshells, chitosan, rice starch, banana peels, moringa seeds, and other nanoparticles are gaining interest in wastewater treatment settings [17,24]. This process is seen as very appealing due to the high efficiency, ability to reduce sludge volume, rapid sedimentation, and low cost [1,5].

Natural coagulants having positively charged functional groups, such as chitosan (CS), have a greater potential for interacting with hydroxyl groups (OH-) in wastewater than cellulose [15,25]. CS is a nontoxic linear poly-amino-saccharide produced by chitin deacetylation and is the second most abundant biopolymer after cellulose [25,26]. CS has a high density, is biodegradable, simple to handle, non-toxic, leaves no residual metal in treated water, and eliminates secondary contamination issues [10,15]. The usage of chitosan produced from bio-waste also creates bigger flocs, allowing for simpler removal while reducing environmental responsibility for manufacturers of crustacean waste products [14,27]. Chitosan’s coagulant activity is highly influenced by the concentration and the pH of the solution [15]. As a result, optimizing operational parameters is important for successful pollution removal. Starch is one of the most abundant polysaccharides on the planet, is inexpensive, and is composed of two anhydroglucose units: linear amylose and highly branching amylopectin [28,29]. The use of starch as a coagulant has sparked a lot of interest in the reduction of turbidity from industrial effluents [30,31]. When starch comes into contact with hot water, the crystalline structure is disturbed, causing permanent swelling of the granules and an increase in viscosity [29,32]. Therefore, this study aims to assess and compare the efficacy of magnetized chitosan and rice starch synthesized by co-precipitation in three different ratios (1:2, 1:1, and 2:1) of natural coagulants (chitosan or rice starch) and magnetite (Fe_3_O_4_) nanoparticles in a magnetized coagulation system for wastewater treatment. 

## 2. Materials and Methods

### 2.1. Material

Sigma-Aldrich (Kempton Park, South Africa), provided the sodium hydroxide (NaOH), ferrous sulphate heptahydrate (Fe^2+^), ferric chloride hexahydrate (Fe^3+^), oleic acid, ethanol, and chitosan (with a deacetylation level of 75–85%) needed in the synthesis of magnetized coagulants. The materials were utilized without being further purified. The rice starch was acquired from a local grocery store (Shoprite, Durban, South Africa), and powdered according to test technique by Sibiya et al. [5] and Pritchard et al. [33]. Deionized water (ELGA WATERLAB, PURELAB Option-Q water deionizer (High Wycombe, UK)) was used for the preparation of the stock solutions.

### 2.2. Wastewater Samples

Wastewater samples were collected after chlorination process at a local wastewater treatment facility in KwaZulu-Natal, South Africa. The samples were characterized according to APHA [34]. Table 1 shows the characterization of the wastewater used in this study. A turbidimeter (Hach, 2100N, Loveland, CO, USA) was used to analyse turbidity, while a spectrophotometer (HACH, DR 3900, Düsseldorf, Germany) was used to analyse color, phosphate, COD, TSS, and absorbance at a wavelength of 465 nm, 880 nm, 620 nm, 810 nm and 620 nm, respectively.

### 2.3. Synthesis of Magnetized Coagulants 

To begin, 1 L stock solutions were calculated for each chemical using their mass (g) and molecular weights, as indicated in Table 2. In reality, the 0.4 M Fe^3+^, 0.2 M Fe^2+^, and 3 M NaOH stock solutions were made using 108.12 g, 55.61 g, and 199.99 g dissolved in 1 L distilled water, respectively [35]. These were calculated using the Sigma-Aldrich mass molarity calculator with a known concentration and volume. 

Figure 1 shows the modified schematic diagram of synthesizing the magnetized coagulants. The Fe_3_O_4_ nanoparticles (F) were synthesized using the co-precipitation technique with a molar ratio of 1:2 for Fe (II): Fe (III) [32,36]. This was then dissolved in 1 L of deionized water with gentle mixing at 30 rpm for 1 h [18]. Following that, 2 mL of oleic acid was added as a surfactant, followed by 2 h of slow mixing (50 rpm) at 70 ± 5 °C [35]. Then, 250 mL of 3 M NaOH was added dropwise to the mixture to correct the pH (11) until a thick black precipitate developed while constantly stirring at 50 rpm. The black precipitate was allowed to cool before being filtered and rinsed with deionized water and ethanol. Thereafter, it was oven-dried at 80 °C for 24 h. The weight ratios of natural coagulant (chitosan or rice starch) to Fe_3_O_4_ were 1:1, 1:2 and 2:1, as shown in Table 3. The samples were well-mixed before being calcined (Kiln Furnace (Cape Town, South Africa)) at 550 °C for 1 h to remove organic contaminants. The MC samples were then packaged for characterization.
(1)$${\mathrm{FeCl}}_{2}\cdot 4H_{2}O+2{\mathrm{FeCl}}_{3}+8\mathrm{NaOH}={\mathrm{Fe}}_{3}O_{4}+8\mathrm{NaCl}+20H_{2}O$$

### 2.4. Characterization of MC

#### 2.4.1. Morphological and Elemental Examination (SEM/EDX)

The elemental analysis of the natural polymers, magnetite and MC were identified using scanning electron microscopy and energy-dispersive X-ray (SEM/EDX) at the University of Cape Town, South Africa. This was accomplished with a 5 kV acceleration voltage and a magnification range of 10 to 50,000x.

#### 2.4.2. Crystal Structure Analysis (XRD)

At 40 kV and a target current of 40 mA, the crystal structures of the generated MC were investigated using an X-ray diffractometer (Bruker AXS, D8 Advance (Madison, WI, USA) coupled with PANalytical software (Malvern PANalytical Ltd., Empyrean, Malvern, UK). The measurements were taken between 15 and 80 (2θ), with a typical step size speed of 0.0340°/min. Lyn-Eye, a position-sensitive detector, was used to gather diffraction data at a typical speed of 0.5 s/step, which was similar to a dazzling counter’s effective length of 92 s/step. The MC diameters were calculated from the XRD data using the Debye–Scherrer Equation (2), which establishes a relationship between particle size and peak enlargement:(2)$$d=\frac{0.98\delta }{\beta Cos\theta }$$
where *d* is the particle size of the crystal, *δ* is the wavelength of X-ray radiation (CuKα = 0.15406 nm), 0.98 is the Scherrer constant, *β* is the line broadening in radians obtained from the full width at half maximum height (FWHM) of the peak (determined using Origin software), and *θ* is the Bragg diffracting angle of the XRD diffraction patterns.

#### 2.4.3. Surface Area Analysis (BET)

The Brunauer–Emmett–Teller (BET) analysis was performed on the equipment (Micromeritics Instrument Corporation, TriStar II Plus version 3.01, Roodepoort, South Africa). The carrier gases were helium and nitrogen (Afrox, Durban, South Africa). Samples were weighed, placed in a sample container on the analyzer, and degassed individually for 24 h at 400 °C. They were then allowed to cool before being stored under nitrogen gas at a pressure of 5 mmHg for 24 h.

#### 2.4.4. Functional and Molecular Examination (FTIR)

The organic, polymeric, and inorganic molecular structures and functional groups of the NPs were recorded using a Fourier transform infrared spectrometer (Shimadzu Corporation, FTIR 8400, Kyoto, Japan) with a resolution of 7 cm^−1^ in the 400–1200 cm^−1^ range.

### 2.5. Coagulation

The coagulation tests were carried out in beakers with 500 mL of wastewater with the use of a jar-test (VELP Scientifica, JTL6, Usmate Velate, MB, Italy) coupled with six paddles [5]. Each beaker was dosed with 4 g of MC (Table 1). Firstly, the types of coagulants were evaluated at a consistent settling time of 60 min. The effect of settling time (10–60 min) on turbidity, color, phosphate, and absorbance removal was also studied. Thereafter, the beakers were agitated at a fast-mixing rate of 150 rpm for 2 min, followed by flocculation at 30 rpm for 15 min [5,18], and then allowed to settle. The effluents were immediately analyzed for turbidity, color, and phosphate in accordance with APHA [34]. The proportion of pollutants (turbidity, color and phosphate) removed were determined by Equation (3):(3)$$Z\left(\%\right)=\frac{C_{a}-C_{b}}{C_{a}}\times 100$$
where $C_{a}$ and $C_{b}$ are the initial and final values of each contaminant (phosphate, turbidity or color) and *Z* is the percentage removal.

## 3. Results and Discussions

### 3.1. Characterization

#### 3.1.1. Elemental Composition by Energy Disperse X-Ray (EDX) and SEM Analysis

Table 4 shows the EDX analyses of magnetite, chitosan, and rice starch, together with their elemental distributions at 20 KeV. The magnetite indicated the following composition: Fe (39.13 percent) > O (38.43 percent) > C (10.36 percent) > S (9.72 percent) > Cl (2.37 percent), as shown in a previous study [5]. The presence of carbon could be due to the doping with carbon gas during the analysis. The composition of chitosan was C (92.89 percent) > O (6.86 percent) > Na (0.25 percent), whereas the composition of rice starch was C (84.55 percent) > O (14.62 percent) > P (0.43 percent) > K (0.40 percent).

Table 5 shows the EDX results for all magnetized coagulants (MCs). Carbon (30.23 to 64.31 percent C), oxygen (22.67 to 43 percent O), sulphur (0.82 to 13.1 percent S), chlorine (0.80 to 15.68 percent Cl), and iron (7.04 to 67.32 percent Fe) were also shown to be the most prevalent elements in all MCs. This is consistent with Equation (1), which demonstrated that magnetite is predominantly composed of Fe and O, while the addition of extra components (S and Cl) may enhance their surface area, allowing for improved biosorption and reusability [35]. In addition to the elements listed above, the RF and CF included potassium (0.35 to 2.14 percent K) and sodium (1.67 to 6.37 percent Na), respectively.

Figure 2a–c depicts the SEM findings of chitosan (CS), rice starch (RS), and magnetite (Fe_3_O_4_) at width distances (WD) ranging from a low to large porosity of 5.30, 5.6, and 6.0 mm, respectively. Magnetite and CS were obtained at a microscale of 1 µm using a magnification of 50 kx, a view field of 4.5 µm, and an accelerating voltage of 5 kV. The RS (Figure 2c) was acquired at a 10 µm microscale, with a magnification of 10 kx, a horizontal view width of 29.8 m, and a landing energy capacity of 20 keV. The change in appearance and view width might be attributed to the 550 °C calcination temperature, which improved the liquid–solid adsorption capability [5,35]. Figure 2a depicts magnetite with a regular cellular appearance and a slightly agglomerated structure [37,38,39], indicating that the iron crystals were partially obscured by carbon particles, possibly due to the sample’s calcination process and carbon coating prior to analysis [35,40,41]. Figure 2b displays the surface aggregation of CS particles, which was heterogeneous and resulted in a smooth surface [42,43]. Figure 2c illustrates RS’s SEM, which had a polygonal irregular form with a smooth surface, as shown in the photos observed by Han et al. [44]. In this work, RS was calcined at temperatures considerably above the temperature (62 to 78 °C) at which crystals shred their deformability, also known as the gelatinization temperature [45]. As a result of the continuous heating (1 h calcination), the RS granules expanded and the crystallites melted, and amylose and amylopectin were entirely separated from the starch [46]. 

Composite surface images (Figure 3a–f) were captured at magnifications of 10× and 50× and a microscale of 5 µm to demonstrate the transformation of un-magnetized coagulants into magnetized coagulants. These binary composites, at various ratios, were seen to have a rough surface. The magnetized chitosan in various ratios (Figure 3a–c) has a compact surface that exhibits a light aggregate, and more pores are visible, particularly in Figure 3c. Similar findings were drawn from Figure 3d–f, the last of which revealed a porous surface with a clearly defined porous structure. The modifications might be the result of calcination at a high temperature (550 °C), which has been observed to alter crystal structures and increase porosity.

#### 3.1.2. Crystal Structure Analysis (XRD)

XRD results were attained in a powdered form of the coagulant as explained in Section 2.4.2. Figure 4 depicts the XRD results for (a) chitosan and its magnetized coagulants, and (b) rice starch and its magnetized coagulants. The XRD pattern of the magnetite produced is very comparable to the JCPDS pattern (Table 6). Figure 4 shows that the reflection at 2θ values of 21.398°, 35.423°, and 46.8° corresponded to the lattice planes of (14), (227), and (148), respectively, for the magnetite. To calculate the size of magnetite crystallites formed at various times, Scherer’s equations with a pseudo-Voigt function were applied to the peak at 2θ of 35.423° [47,48]. 

The examined diffraction peaks matched normal magnetite XRD patterns more closely than maghemite, indicating a cubic crystalline structure [49]. Figure 4a depicts the XRD patterns of CS and its magnetic coagulants. The chitosan pattern exhibited one unique broad diffraction peak at 2θ = 20.15°, which is characteristic of semi-crystalline chitosan fingerprints [50,51,52,53]. This peak in the chitosan structure is related to crystal I and crystal II, and it suggests that the chitosan has a high degree of crystallinity [43,54]. The (225) plane corresponds to the apex at around 30.1°. Its magnetized coagulants behave similarly, with a noticeable peak at 35.423°, which corresponds to the (227) plane. The XRD image of rice starch and its magnetized coagulants is shown in Figure 4b. Rice starch exhibited a characteristic A-type XRD pattern, with prominent peaks at 2θ = 17.374°, 24.482°, and 25.432°, that were universally acknowledged as the usual cereal starch and legume crystal shape, which corresponds to the lattice patterns of (160), (225), and (63), respectively [55,56,57,58]. The wide and difficult-to-distinguish diffraction peaks revealed the presence of very tiny crystallite sizes [58]. 

Starches with an A-type diffraction pattern are said to have a helical arrangement in monoclinic symmetry unit cells [59]. RF (2:1), RF (1:1), and RF (1:2) demonstrated a similar pattern, with large peaks at 33.153°, 35.423°, 54.5°, and 63.5°, which correspond to the lattice planes of (104), (227), (116), and (214), respectively, and are identical to Matmin et al. [58]. Figure 4b shows that RF (1:1) has greater intensity values than the other RF ratios.

#### 3.1.3. Surface Area Analysis (BET)

Table 7 displays the BET results. Due to magnetite’s specific magnetic properties and durability, integrating Fe_3_O_4_ into natural coagulants (chitosan or rice starch) enhanced the surface area of the magnetic coagulants, enhancing their adsorption capability [5,30,35]. Amongst the coagulants examined, RF (1:1) exhibited the highest surface area of pores and porosity. Surface area is shown in descending order: RF (1:1) > RF (2:1) > RF (1:2) > Magnetite> CF (1:1) > CF (2:1) > CF (1:2) > Rice starch > Chitosan. Mosaddegh et al. [60] state that the bigger the surface area of coagulant particles, the higher the reaction conversion.

#### 3.1.4. FTIR Analysis 

FTIR spectroscopy was used to determine the functional groups of the coagulants studied. The 200 scans were obtained between 400 and 1200 cm^−1^ with a resolution of 7 cm^−1^ for each measurement (Figure 5). The peaks between 1200 and 900 cm^−1^ correspond to the vibrational region in concordance with the surfactant (oleic acid) employed in this study [61]. The absorption bands of the magnetite were observed at 1055, 936, 846, 548 and 436 cm^−1^ (Figure 5). The C–O stretching band, which is connected with the C–O–SO_3_ group, was responsible for the peak at 1055 cm^−1^ [62]. The appearance of an aromatic C–H bending band was revealed by the absorption peaks 936 and 846 cm ^−1^. Magnetite also had two significant peaks at 548 and 436 cm^−1^, which correspond to the stretched vibration mode of Fe–O [63]. The metal–oxygen band at 436 cm^−1^ was ascribed to octahedral-metal stretching of Fe–O in magnetite, whereas the metal–oxygen band at 548 cm^−1^ was given to intrinsic metal stretching vibrations at the tetrahedral site [49]. These distinctive peaks proved the development of iron oxide nanoparticles, since the peaks positioned between 400 and 600 cm^−1^ matched to the magnetite phase [64].

Figure 5a depicts absorption bands typical of the chitosan saccharine structure at 1160 cm^−1^ (asymmetric stretching of the –COOC– bridge), 1086 cm^−1^, and 1030 cm^−1^ (skeletal vibration involving the COO stretching) [52]. Off-plane bending –OH vibrations are also recorded at 680 cm^−1^. In general, the FTIR spectra of chitosan show discrete bands corresponding to amide groups between 1305 and 1660 cm^−1^ [65,66], N-H peaks, and hydrogen-bonded hydroxyl groups between 3270 and 3800 cm^−1^ [67]. The FTIR spectra of RS and its magnetic coagulants (Figure 5b) revealed a particularly strong peak at about 995 cm^−1^, which was attributed to the C–O of starch’s C–O–C bonds [68]. It also reveals that shoulder peaks at around 939 cm^−1^ (skeletal mode vibrations of −1.4 glycosidic linkage), 1085 cm^−1^ (C–O–H bending), and 1148 cm^−1^ (C–O, O–H, and C–C stretching) were seen, comparable to the studies by Talekar et al. [69] and Fan et al. [70]. C–H and CH_2_ deformation created a noticeable peak at 852 cm^−1^ in the region below 900 cm^−1^ [30]. The C–C stretching produced a peak at 760 cm^−1^, while peaks observed between 600 and 400 cm^−1^ were discovered to represent skeletal modes of pyranose [58].

Changes in the distinctive spectra peaks altered the physical versus chemical interactions when rice starch was combined with magnetite [51]. The RFs behaved similarly throughout, however the RF (1:2) had the maximum transmittance > RF (2:1) > RF (1:1). (1:1). In RF spectrums, the C–O stretch peak changed from 1148 to 1100 cm^−1^. The C–C stretch peak also moved from 852 to 689 cm^−1^. This occurrence demonstrated the presence of interactions between the hydroxyl groups of RS and magnetite, which may result in the formation of hydrogen bonds [51]. 

### 3.2. Evaluation of Coagulant Types

Figure 6 shows the turbidity removal efficiencies of the magnetic coagulants. According to the results, the RF reduced turbidity at a rate of more than 75%, ranging from 80% to 85%, compared to 71–74.27% for the CF. The amine functional groups in chitosan, as well as the positively charged magnetite, aided the coagulation and adsorption of negatively charged pollutants in the wastewater sample [10]. The mechanism of cationic exchange (electrostatic, van der Waals, and chemical bonding) and the generation of hydroxide on the surface of magnetized rice starch coagulants was used to remove turbidity by magnetic means [18,32,71]. Furthermore, amylase and amylopectin from rice starch were used to destabilize colloidal particles by bridging and aggregation [32]. The decreasing sequence of turbidity removal was seen to be RF (1:1) > RF (1:2) > RF (2:1) > CF (1:1) > CF (1:2) > CF (2:1), with efficiencies as of 84.81% > 82.07% > 74.27% > 73.54% > 71.85%, respectively. Rice starch has been identified as one of the most effective coagulants for turbidity reduction [72], and due to the concentration of cationic elements, it is suitable for the adsorption of negatively charged particles [73,74].

Other contaminants that were reported for all the coagulants are displayed in Table 8. For each coagulant, the percentage removal results for phosphate, color, and absorbance were recorded. For all contaminants, RF (1:1) attained the maximum phosphate, color, and absorbance removal efficiencies of 92.98, 82.54, and 95.58%, respectively, which affirms with the BET results (Table 7) that surface characteristics aid in the performance of the coagulants, hence high surface area with high porosity will impact the process highly as compared to the lower surface area.

### 3.3. Evaluation of Settling Time on Contaminants Removal

Settling time is one of the most important parameters to consider for the coagulation and flocculation process, since it affects the total cost and efficiency of the coagulation process [74]. Figure 7a depicts color removal efficiency, which indicates a rise throughout the lag period (10–20 min). The results demonstrated rapid initial removal, which was explained by the availability of unoccupied sites as the adsorbent was more porous and had a large surface area to adsorb more contaminants, and due to the energy and time constraints, a maximum color removal efficiency at 20 min (84.76%) is preferred. A similar observation was reported by Sibiya et al. [5] and Sun et al. [75]. Furthermore, the shorter the settling period, the better the process in terms of sludge reduction or handling [76,77]. Increasing the settling time beyond 20 min, on the other hand, does not appear to be effective for color removal (dropping to 80% at 40 min), showing that colloidal particles are destabilized as a result of charge reversal. The color removal efficiency rose from 50 to 60 min (84.13%), but there was no improvement beyond that due to increasing settling time causing a limited quantity of flocs clots to develop and be deposited by gravity [78]. 

Figure 7b depicts the phosphate removal efficiency over a period of 60 min settling time and an increase from 10 (86.44%)–30 min (95.76%) was observed but a decrease on the 40th min of settling, and then an increase at the 60th min. This irregular pattern may be due to sampling as the settling process continues, causing the trapped pollutant to break apart and increase the amount of phosphate. This also confirms the drawbacks of the coagulation process’s settling time. Other studies have shown that low effectiveness might be caused by pH, because at low pH, the coagulant’s surface is heavily protonated by positive ions (H^+^), resulting in an electrostatic strength between the contaminated molecules and the surface, resulting in strong adsorption [79,80]. Phosphate removal in wastewater is significant because high phosphate wastewater is dangerous, since it promotes the growth of poisonous bacteria and algae by providing nutrients to aquatic organisms. Furthermore, it degrades the water’s aesthetics and reduces the quantity of soluble oxygen in the water. In this vein, the removal of phosphate from wastewater is compulsory before it is discharged into river bodies.

Figure 8a depicts the turbidity efficiencies for the best coagulant (RF (1:1)). The results showed that increasing the settling time (10 to 60 min) enhanced the efficacy of removal until the time reached 40 min, at which point the effectiveness begins to decrease. The coagulant provides the turbidity removal above 80% at all the settling times. The residual turbidity diminishes between 10 and 40 min due to the formation of large-size flocs with a higher settling velocity [74,78], while from 40 to 50 min, the trend is almost reversed. As the settling period increases from 50 to 60 min, the relevance of the sweep coagulation mechanism outweighs that of the charge neutralization mechanism due to the high degree of super-saturation [13,78]. 

The modest flocs growth rate in comparison to the flocs nucleation rate is most likely the cause of efficient particle bridging; polymer chains have a greater inclination to overlap with neighboring polymer chains [81]. Figure 8b depicts the COD removal efficiency after 60 min of settling. The findings show a rising tendency from 10 (50.65%) to 30 min (55.04%). After 30 min, an uneven pattern is shown, which might be due to sampling during the settling phase, which caused the contained pollutant to break apart and increase the amount of COD. This also confirms the drawbacks of the coagulation process’s settling time. Other research has proposed that limited efficiency may be linked to pH, since at low pH, the coagulant’s surface is massively protonated by cations (H+), resulting in an electrostatic strength between the contaminated molecules and the surface, generating high adsorption [16]. This finding demonstrates that coagulation with natural coagulants, particularly rice starch, was beneficial in treating wastewater [30,32]. Other researchers had used a similar technique to prove the efficacy of wastewater treatment equipment [82].

### 3.4. Comparison of Several Magnetized Coagulants Used in Wastewater Treatment

When the efficacy of coagulants identified in the literature is compared, natural coagulants (such as eggshells, chitosan, and rice starch) functionalized with magnetite show significant promise for possible large-scale applications in industrial wastewater treatment, as shown in Table 9.

## 4. Conclusions

This study describes advances in the development of an ecologically friendly water-treatment system based on magnetic iron oxide nanoparticles and their functionalization with rice starch and chitosan. The magnetic coagulants (chitosan and rice starch) were evaluated to determine the best concentration of each natural coagulant and iron oxide nanoparticles (ratios) in the analyzed systems. The analytical findings from the Brunauer–Emmett–Teller (BET) analyzer, X-ray diffraction (XRD) analyzer, and energy-dispersive X-ray (EDX) spectroscopy confirmed the success of the magnetized coagulants’ (MCs) surface area, crystal structure, and elemental compositions, respectively. According to the BET results, introducing magnetite to natural coagulants significantly increased the pore size of the MCs. Amongst the coagulants, RF (1:1) efficiently removed 84.81% turbidity, 82.54% color, 92.98% phosphate and 95.58% absorbance. Therefore, a combination of natural coagulant rice starch and magnetic nanoparticles improved the system’s ability to coagulate contaminants in the effluent. For the best coagulant, the impact of settling time (10–60 min) at a consistent dose of 4 g was studied. The results revealed that RF (1:1) eliminated turbidity (88.16%), COD (55.04%) and phosphate (95.76%) after 30 min, and color (84.76%) after 20 min. The system’s capacity to coagulate impurities was enhanced by functionalizing rice starch and magnetite. The RF (1:1) synthesized in this work is a fascinating approach with the potential to deliver a cost-effective, resilient, and environmentally friendly water treatment technology.

## Figures and Tables

**Figure 1 polymers-14-04342-f001:**
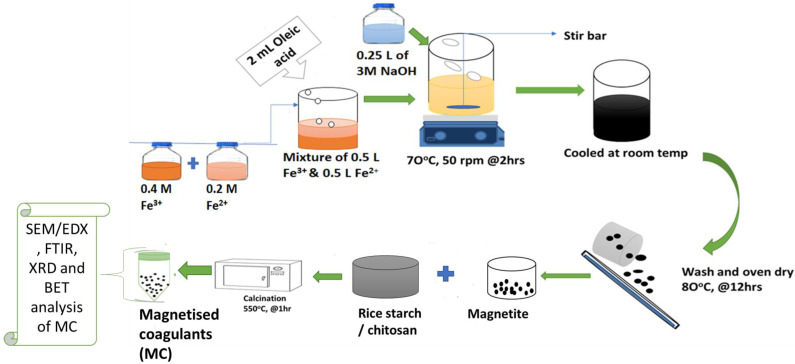
Schematic diagram of synthesizing magnetic coagulants.

**Figure 2 polymers-14-04342-f002:**
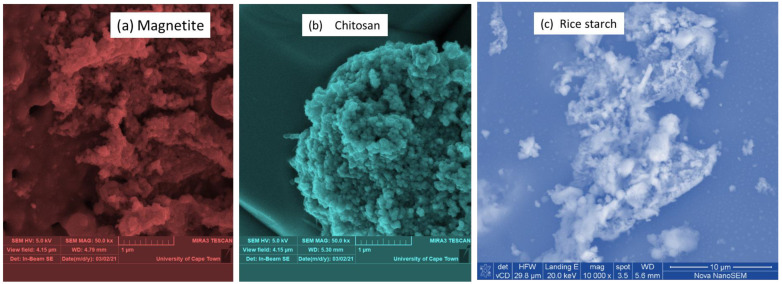
SEM images (**a**) magnetite, (**b**) chitosan, and (**c**) rice starch.

**Figure 3 polymers-14-04342-f003:**
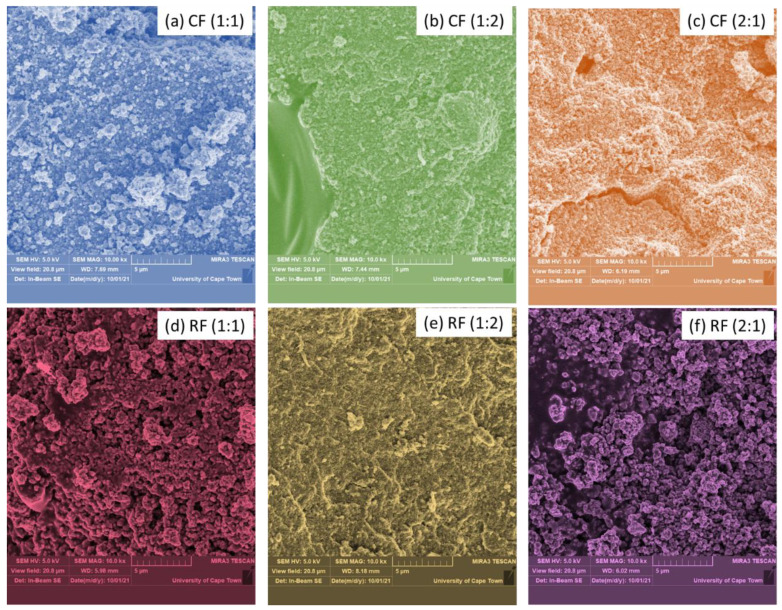
SEM images (**a**–**f**) of the magnetized coagulants.

**Figure 4 polymers-14-04342-f004:**
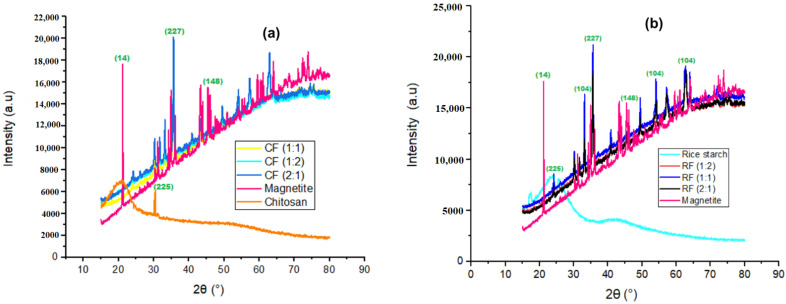
XRD spectra (**a**), Chitosan and its magnetized coagulants (**b**) Rice starch and its magnetized coagulants.

**Figure 5 polymers-14-04342-f005:**
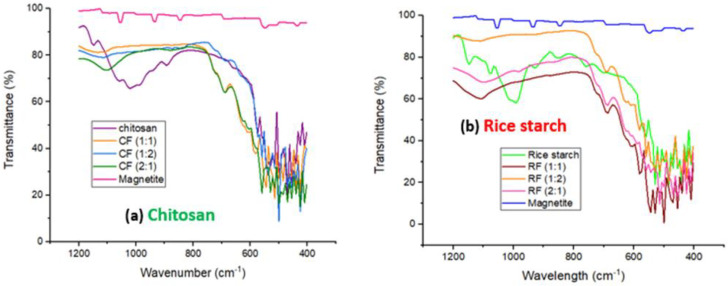
FTIR spectra (**a**) Chitosan and its magnetized coagulants; (**b**) Rice starch and its magnetized coagulants.

**Figure 6 polymers-14-04342-f006:**
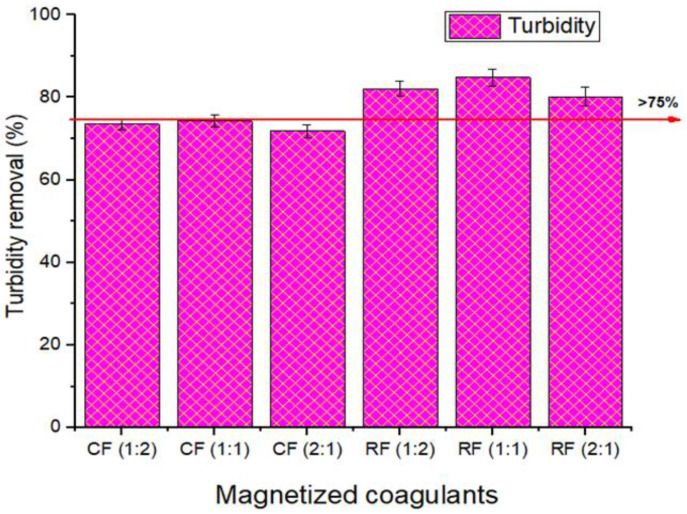
Evaluation of different coagulants on turbidity removal (%).

**Figure 7 polymers-14-04342-f007:**
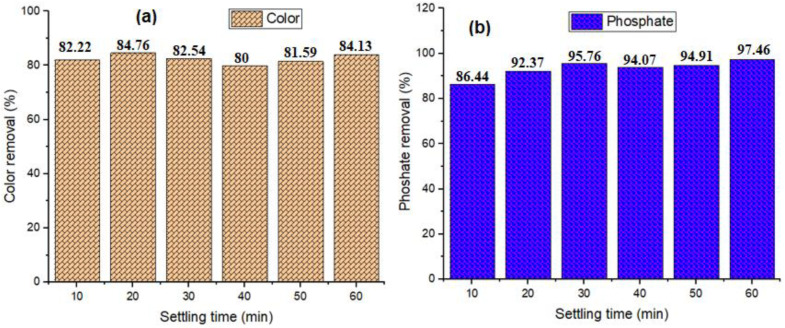
Effect of settling time using RF (1:1) for color removal (**a**) and phosphate removal (**b**).

**Figure 8 polymers-14-04342-f008:**
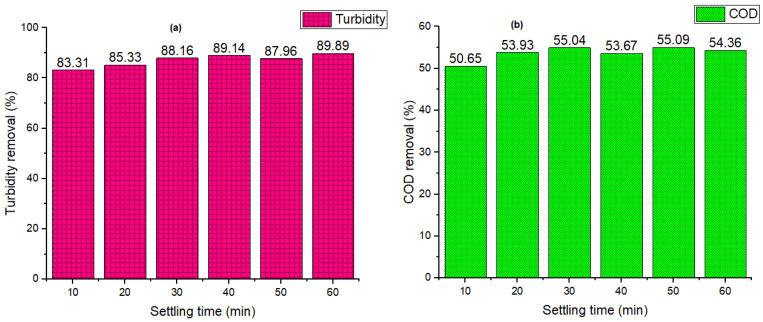
Evaluation of different coagulants on (**a**) turbidity removal (%), and (**b**) COD removal (%).

**Table 1 polymers-14-04342-t001:** Characteristics of wastewater sample composition.

Contaminants	Values	Standard Deviation
Turidity (NTU)	45.60	0.2910
Color (Pt. Co)	315	0.098
Absorbance (%)	73.40	0.151
Chemical oxgyen demand (COD) (mg/L)	352	0.816
Total suspended solids (TSS) (mg/L)	68.20	0.748

**Table 2 polymers-14-04342-t002:** Stock solution composition.

Chemicals	Concentration (M)	Molar Mass (g/mol)	Mass (g)
FeCl_3_ 6H_2_O	0.4	270.29	108.21
FeSO_4_ 7H_2_O	0.2	287.55	55.61
NaOH	3	39.997	199.99

**Table 3 polymers-14-04342-t003:** Amounts measured for natural coagulant and magnetite to prepare 10 g of magnetized coagulants.

Ratios of Magnetized Coagulants (MC)	CF (g)		RF (g)	
	CS (g)	F (g)	R (g)	F (g)
1:1	5	5	5	5
2:1	6.667	3.333	6.667	3.333
1:2	3.333	6.667	3.333	6.667

(CF—magnetized chitosan and RF—magnetized rice starch).

**Table 4 polymers-14-04342-t004:** EDX results for the magnetite, chitosan, and rice starch.

Elements	C	O	S	Fe	Na	P	K	Cl	Ca
Weight	%	%	%	%	%	%	%	%	%
Magnetite	10.36 ± 6.90	38.43 ± 17.12	9.72 ± 3.31	39.13 ± 14.39	-	-	-	2.37 ± 1.64	
Chitosan	92.89 ± 9.73	6.86 ± 9.39	-	-	0.25 ± 0.34	-	-	-	-
Rice starch	84.55 ± 1.68	14.62 ± 1.91	-	-	-	0.43 ± 0.29	0.40 ± 0.15	-	-

**Table 5 polymers-14-04342-t005:** EDX results for the all magnetized coagulants.

Elements	C	O	S	Fe	Na	P	K	Cl	Ca
Weight	%	%	%	%	%	%	%	%	%
CF (1:2)	-	27.12 ± 13.15	-	67.32 ± 18.88	3.36 ± 4.29	-	-	2.20 ± 3.09	
CF (1:1)	-	22.67 ± 14.13	-	74.80 ± 15.81	1.67 ± 1.21	-	-	0.80 ± 1.18	0.07 ± 0.16
CF (2:1)	-	32.83 ± 8.13	0.82 ± 0.64	57.44 ± 14.2	6.37 ± 4.83	-	-	2.54 ± 2.32	
RF (1:2)	-	42.81 ± 8.99	13.10 ± 2.60	26.32 ± 5.61	-	-	2.14 ± 0.38	15.68 ± 1.92	-
RF (1:1)	30.23 ± 13.32	34.44 ± 15.87	5.54 ± 3.74	23.34 ± 11.98	-	-	2.14 ± 0.38	6.44 ± 5.29	-
RF (2:1)	64.31 ± 15.02	24.48 ± 5.40	0.83 ± 0.25	7.04 ± 8.24	-	-	0.55 ± 0.27	2.99 ± 2.76	-

**Table 6 polymers-14-04342-t006:** Physical and chemical properties of the MCs characterized obtained from the XRD.

2θ (Degree)	Miller Indices Plane (hkl)	dhkl (nm)	Crystal Structure	Nanostructure	JCPDS Pattern
24.482	(225)	1.988	Face-centered cubic	Sylvite	00-41-1476
46.8	(148)	3.157	Rhombohedral	Mikasaite	00-047-1774
21.398	(14)	3.202	Monoclinic	Ferrimagnetite	00-070-2091
48.491	(12)	2.106	Monoclinic	Clinoptilolite	01-071-1425
51.44	(44)	2.162	Base-centered Orthohombic	Sodium nitrate	01-075-2073
27.335	(225)	2.163	Face-centered cubic	Halite (NaCl)	00-005-0628
32.497	(29)	2.263	Orthohombic	Thermonatrite	00-008-0448
20.212	(13)	1.96	Monoclinic	Iron chloride hydrate	00-016-0123
35.525	(227)	4.523	Face-centered cubic	Magnesioferrite	00-017-0464
35.423	(227)	5.197	Face-centered cubic	Magnetite (Fe_3_O_4_)	00-019-0629
17.374	(160)	3.065	Rhombohedral	Hydronium jarosite	00-031-0650
33.153	(104)	5.27	Rhombohedral	Hematite (α-Fe_2_O_3_)	00-033-0664
24.717	(148)	3.06	Rhombohedral	Mikasaite	00-033-0679
35.631	(110)	4.858	Cubic	Maghemite (y-Fe_2_O_3_)	00-039-1346

**Table 7 polymers-14-04342-t007:** Comparison of the BET surface area of the coagulants.

Sample/(s)	S_BET_ Surface Area (m^2^/g)	Pore Volume (cm^3^/g)	Pore Size (nm)
Magnetite	27.597	0.0080	1.4840
Rice starch	1.267	0.0020	6.7600
RF (1:1)	31.438	0.0015	1.6102
RF (2:1)	30.021	0.0012	1.6098
RF (1:2)	29.388	0.0010	1.5418
Chitosan	1.2010	0.0007	5.4180
CF (1:1)	18.773	0.0008	4.5560
CF (2:1)	16.291	0.0008	4.5110
CF (1:2)	13.918	0.0004	3.7361

**Table 8 polymers-14-04342-t008:** Comparison results of different types of coagulants.

Contaminant	Removal Percentage (%)					
	CF (1:1)	CF (1:2)	CF (2:1)	RF (1:1)	RF (1:2)	RF (2:1)
Phosphate	86.16	81.92	79.86	92.98	90.94	88.83
Color	76.30	76.09	74.73	82.54	79.05	80.69
Absorbance	92.97	94.72	92.27	95.58	94.97	94.37

**Table 9 polymers-14-04342-t009:** Comparative study on various nanocomposites used in wastewater treatment.

Magnetized Coagulant	Operating Conditions	Contaminants Removal (%)	Reference
Magnetized moringa oleifera	30 min settling time	68.33% color	[83]
Magnetized alum	50 min magnetic exposure	85% turbidity 82% color	[18]
Magnetized chitosan	pH = 6 370 mg/L dosage	97.7% TSS 91.70% COD 92.70% turbidity	[10]
Magnetised rice starch	17.33 settling time 3.40 g dosage	69.96% turbidity 45.51% phosphate	[82]
RF (1:1)	20 min settling time 4 g dosage	84.76% color 53.93% COD 92.37% phosphate 85.33% turbidity	This study
Magnetized eggshell	30 min settling time	94.86% TSS 92.56% turbidity 96.24% color	[5]

## Data Availability

Not applicable.

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
