# Peer review of "Effect of Magnetized Coagulants on Wastewater Treatment: Rice Starch and Chitosan Ratios Evaluation"

_polymers, 2022, doi:10.3390/polym14204342_

Round 1
Reviewer 1 Report
The reviewed paper entitled ‘Effect of magnetized coagulants on wastewater treatment: A comparative study” is well written and describes interesting research. However, while the characterization of these materials is enough to imply their structural characteristics, more studies are needed to fully understand their removal performance. Without both a detailed study on the removal capabilities of these materials and evidence that the synthetic control they have developed can be used to enhance this capability, it is difficult to justify the acceptance of this manuscript to this journal. Overall, this manuscript presents an important and potentially impactful contribution to the existing literature on magnetized coagulants which is recommended for publication in the Journal Polymers following the major revisions detailed below.
Prior to publication the authors should address the following major issues.
- The authors should provide the characterization of the materials after the contaminant’s removal. What happens to the BET surface area, EDS analysis, etc.
- The use of the term “comparative study” in the title is misleading, as well as the last sentence in Introduction Section. Based on literature, the purpose of this research is to assess the efficacy of magnetized chitosan and rice starch made by co-precipitation in three different ratios (1:2, 1:1, and 2:1) of natural coagulants (chitosan or rice starch) and magnetite (Fe3O4) nanoparticles in a magnetized coagulation system for wastewater treatment.
These expressions foreshadow the reader for a review study and not for an experimental article.
3. Which is the importance to decrease the color of the wastewater if the color still exists (even in a percentage of 15%). There are similar concerns for the other two pollutants, i.e. turbidity and phosphate. Are there specific established limits for these pollutants in drinking water (from US-EPA, WHO, EEA)?
4. A paragraph dedicated to the mechanism of the removal in any case should be included in the Results Section.
5. The authors should include a comparison of the performance of their materials with other materials present in the literature to better contextualize their results.
In addition to the above Major revisions, the authors should also address the following minor revisions:
6. Fig 2 and Fig. 3 should be improved. The elements are not clearly presented. Also, in Fig. 2 the authors should improve the panel.
7. In Fig. 4 and in the manuscript: the authors should mention that it is XRD analysis in powder samples, the patterns could be presented as an offset for clarity.
8. Beginning on line 187, the authors state “Furthermore, the presence of carbon (C) due to the samples being covered with carbon gases” What does that this sentence imply? Carbon tape should be taken under consideration in EDX analysis.
9. Did the authors use IR spectroscopy for the characterization of the magnetic coagulants?
Author Response
Thanks very much for the critical comments and suggestions for improving the technicality of the manuscript as presented in the revised manuscript. the document attached addresses all the comments.

Reviewer 2 Report
Reviewer’s comments on the manuscript: Effect of magnetized coagulants on wastewater treatment: A comparative study written by Nomthandazo Precious Sibiya, Gloria Amo-Duodu, Emmanuel Kweinor Tetteh and Sudesh Rathilal
The aim of the reviewed manuscript is to assess the efficacy of magnetized chitosan and rice starch made by co-precipitation in three different ratios (1:2, 1:1, and 2:1) of natural coagulants (chitosan or rice starch) and magnetite (Fe3O4) nanoparticles in a magnetized coagulation system for wastewater treatment. The manuscript presents quite interesting results and in my opinion is in the journal’s fields of interests. However, I have reservations about nomenclature thus some additional information should be added or clarified before publications process. Thus my suggestion is major revision.
Reviewers comments and suggestions:
· All manuscript: I am not convinced to the usage of the term ‘coagulation’ in the case of the proposed measurements. Coagulation occurs in the presence of salts: Al2(SO4)3, Fe2(SO4)3, AlCl3, and FeCl3 but the Authors use high molecular weight compounds: magnetized chitosan and rice starch. Such substances and not coagulants but flocculants. The effects of their influence on wastewater are similar but the mechanisms are completely different. If the flocculant and coagulant are used simultaneously the process is usually called the CFS process (coagulation, flocculation and sedimentation). In the reviewed manuscript polysaccharides are used together with magnetite so I my opinion the terms coagulants and coagulation are not the best terminology in this case. They should be corrected. Please think over this problem and make necessary changes in the manuscript.
· All manuscript: I have no idea why some parts of text were highlighted (yellow). It looks like the previous not final version of the manuscript.
· All manuscript: This's probably the template's fault but often words are not divided correctly into syllables.
· All manuscript: Sometimes there is a space between the number and unit sometimes there is not, please unify this.
· Abstract underlines the relevance of the presented studies.
· Line 14: Could you please explain why such ratios were used?
· Introduction part is well written and informative. Only these coagulants…
· Lines 121, 122: Bolds are not necessary.
· Lines 124, 225, 256, 278: Editorial mistakes.
· Line 142: “a magnification of 10–50 kx”? What do you mean?
· Material characterization: In my opinion one crucial thing is missing here. I mean the molar masses of the used polysaccharides. In the case of high molecular weight substances the molar mass highly influence their properties for example flocculation effectiveness thus in my opinion it is very important to present information on these values.
· Line 193: I completely do not like the way of presentation the EDX data. The figure is curved and hardly legible in fact it looks like a screen. Do you really want to present date in this way? The same with Fig. 3. Maybe tables will be better solution.
· Lines 183-203: The obtained results are clearly presented but the discussion is to weak.
· Line 286: Figure 6. Figure 6.
Author Response

(The authors gave the same response as above.)

Reviewer 3 Report
Comments
Figure 1 is very blur, Fig 2 and Fig. 3 are not clearly visible, improve the quality of all the figures.
Redraw Fig. 2 and Fig 3, without color.
Provide SEM of samples correspond to EDX (Fig 2 and Fig. 3).
You mentioned that results were analyzed on the basis of color, phosphate and absorbance, what is difference between color and absorbance results. Why you selected the wavelength of 465 nm and 620 nm?
Provide spectra of color for real wastewater.
As you mentioned in materials methods, wastewater sample was collected from treatment plant, was the sample treated or untreated?
Page 3, Line 129: Give name of equipment used for calcination of samples. In Fig. 1 calcination device is look like microwave oven.
Provide complete characterization results of real wastewater used in this study (in Table). Must include TSS, TDS, conductivity and COD or BOD.
Page 5, Line 267, clarify this sentence “92.98, 82.54, and 95.58%, respectively, which affirms with BET results (Table 3)”.
Give complete mechanism of flocculation and coagulation process, it will be more attractive if provided in graphical from. In results and discussion portion mechanism provided is literature based, proved mechanism of reaction based on this study.
Provide chemical reactions involved in flocculation and coagulation processes.
What could be the end products after coagulation in case of phosphate?
pH and dose of coagulants are very important parameters in coagulation of wastewater. If you done Jar-test where are the optimized results of pH and dose/concentrations for turbidity removal?
How you selected the 2g dose of coagulants?
In Fig. 7 what is meant by settling time is it reaction time?

Author Response

(The authors gave the same response as above.)

Round 2
Reviewer 1 Report
The authors responded to my comments, providing the missing data/experiments and convincing explanations. Thus, I recommend the publication of this article in polymers in the present form.
Author Response
Thanks very much for the critical comments and suggestions for improving the technicality of the manuscript as presented in the revised manuscript.
Reviewer 2 Report
The provided explanations for my comments are convincing. I advise the acceptance of the manuscript.
Author Response

(The authors gave the same response as above.)

Reviewer 3 Report
In Fig 8b add COD removal % instead of absorbace %.
There are noise in FTIR results (In Fig 5).
Give complete information about sampling point at wastewater treatment facility (in section 2.2).
Author Response

(The authors gave the same response as above.)
